# AMORTIZED POSTERIOR SAMPLING WITH DIFFUSION PRIOR DISTILLATION

## ABSTRACT

We propose Amortized Posterior Sampling (APS), a novel variational inference approach for efficient posterior sampling in inverse problems. Our method trains a conditional flow model to minimize the divergence between the variational distribution and the posterior distribution implicitly defined by the diffusion model. This results in a powerful, amortized sampler capable of generating diverse posterior samples with a single neural function evaluation, generalizing across various measurements. Unlike existing methods, our approach is unsupervised, requires no paired training data, and is applicable to both Euclidean and non-Euclidean domains. We demonstrate its effectiveness on a range of tasks, including image restoration, manifold signal reconstruction, and climate data imputation. APS significantly outperforms existing approaches in computational efficiency while maintaining competitive reconstruction quality, enabling real-time, high-quality solutions to inverse problems across diverse domains.

## 1 INTRODUCTION

We consider the following inverse problem

$$\boldsymbol{y} = \mathcal{A}(\boldsymbol{x}) + \boldsymbol{n}, \quad \boldsymbol{y} \in \mathbb{R}^m, \, \boldsymbol{x} \in \mathbb{R}^n, \, \mathcal{A} : \mathbb{R}^n \mapsto \mathbb{R}^m, \, \boldsymbol{n} \sim \mathcal{N}(0, \sigma_y^2 \boldsymbol{I}), \tag{1}$$

where the goal is to infer an unknown signal $\boldsymbol{x}$ from the degraded measurement $\boldsymbol{y}$ obtained through some forward operator $\mathcal{A}$, leveraging the information contained in the measurement and the prior $p(\boldsymbol{x})$. A powerful modern way to define the prior is through diffusion models (Ho et al., 2020; Song et al., 2021c), where we train a parametrized model $\boldsymbol{s}_\theta$ to estimate the gradient of the log prior $\nabla_{\boldsymbol{x}} \log p(\boldsymbol{x})$.

Solving inverse problems with the diffusion model can be achieved through posterior sampling with Bayesian inference. Arguably the standard way to achieve this is through modifying the reverse diffusion process of diffusion models (Daras et al., 2024). This adjustment shifts the focus from sampling from the trained prior distribution $p_\theta(\boldsymbol{x}_0)$ to sampling from the posterior distribution $p_\theta(\boldsymbol{x}_0|\boldsymbol{y})$. This transition is facilitated by employing iterative projections to the measurement subspace (Kadkhodaie & Simoncelli, 2021; Song et al., 2021c; Chung et al., 2022b; Wang et al., 2023), guiding the samples through gradients pointing towards measurement consistency (Chung et al., 2023a; Song et al., 2023a). It should be noted that diffusion models learn the gradient of the prior, and diffusion samplers (Song et al., 2021a; Lu et al., 2022; Song et al., 2021c) are methods that numerically solve the probability-flow ODE (PF-ODE) that defines the reverse diffusion sampling trajectory. Consequently, regardless of the specifics of the methods, standard diffusion model-based inverse problem solvers (DIS), even those that are considered *fast*, take at least a few tens of neural function evaluation (NFE), making them less effective for time-critical applications such as medical imaging and computational photography.

Another class of methods (Feng et al., 2023; Feng & Bouman, 2023) introduces the use of variational inference (VI) to *train* a new proposal distribution $q_\phi^{\boldsymbol{y}}(\boldsymbol{x})$ to *distill* the prior learned through the pretrained diffusion model. The problem is defined as the following optimization problem

$$\min_\phi D_{KL}(q_\phi^{\boldsymbol{y}}(\boldsymbol{x}_0) || p_\theta(\boldsymbol{x}_0|\boldsymbol{y})), \tag{2}$$

where the superscript $\boldsymbol{y}$ emphasizes that the proposal distribution is specific for a single measurement $\boldsymbol{y}$. For tractable optimization, $q$ is often taken to be a normalizing flow (Rezende & Mohamed,

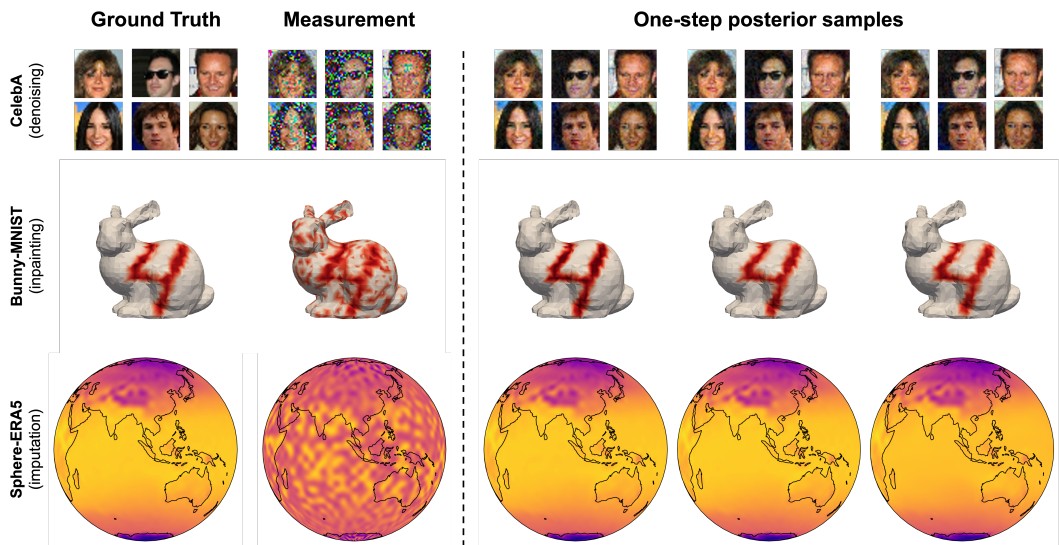

Figure 1: Diverse inverse problem solving can be done with a **single** NFE, with the **same** network for all the different measurements. (row 1) Denoising on celebA, (row 2) inpainting MNIST on the bunny manifold, (row 3) imputation of ERA5 on the spherical manifold.

2015; Dinh et al., 2016) (NF) so that the computation of likelihood can be done instantly, and one can sample multiple reconstructions from the posterior samples by plugging in different noise values from the reference distribution. Posing the problem this way yields a method that can achieve posterior samples with just a single NFE. Nevertheless, it is still impractical as training a measurement-specific variational distribution takes hours of training. It is often unrealistic to train a whole new model from scratch every time when a new measurement is taken.

In this work, we take a step towards a *practical* VI-based posterior sampler by distilling a diffusion model prior. To this end, we propose a *conditional* normalizing flow $q_\phi(x_0|y)$ as our variational distribution and amortize the optimization problem in Eq. (2) over the conditions $y$. By using a network that additionally takes in the condition $y$ as the input, we can train a *single* model that generalizes across the whole dataset without the need for cumbersome re-training for specific measurements. (See Fig. 2 for the conceptual illustration of the proposed method, as well as representative results presented in Fig. 1.) Interestingly, we find that the speed of optimization is not hampered with such amortization, and the proposed method achieves comparable performance against the measurement-specific flow model (Feng & Bouman, 2023; Feng et al., 2023). Furthermore, we extend the theory to consider inverse problems on the Riemannian manifold, showing that the proposed idea is generalizable even when the signal is not one the Euclidean manifold. In summary, our contributions and key takeaways are as follows

1. We propose an amortized variational inference framework to enable 1-step posterior sampling constructed implicitly from the pre-trained diffusion prior $p_\theta(x)$ for *any* measurement $y$.

2. To the best of our knowledge, our method is the first diffusion prior distillation approach for solving inverse problems that are unsupervised (i.e. does not require any ground-truth data $x$), as opposed to standard conditional NFs (Lugmayr et al., 2020) that required supervised paired data.

3. Experimentally, we show that the proposed method easily scales to signals that lie on the standard Euclidean manifold (e.g. images) as well as signals that lie on the Riemannian manifold, achieving strong performance regardless of the representation.

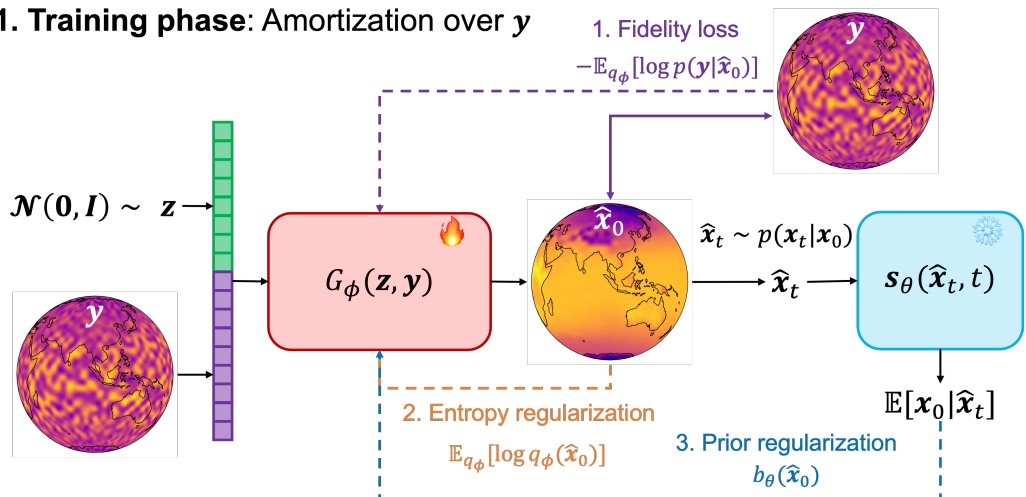

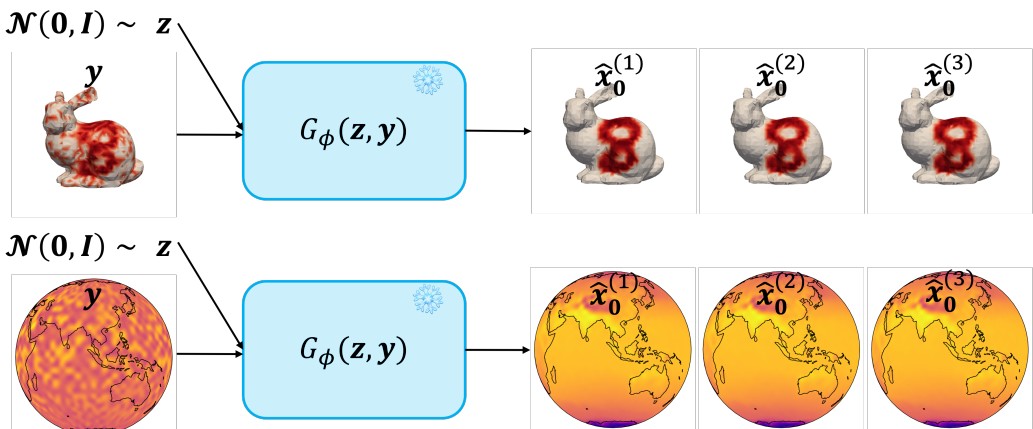

Figure 2: Concept of the proposed method, APS. (a) Training can be performed in an unsupervised fashion with a dataset consisting of degraded measurements $\boldsymbol{y}$ to train a conditional normalizing flow $G_\phi$ with the diffusion prior $\boldsymbol{s}_\theta$. (b) Once trained, one can achieve multiple posterior samples by inputting different noise vectors $\boldsymbol{z} \sim \mathcal{N}(0, \boldsymbol{I})$ concatenated with the condition $\boldsymbol{y}$ with a single NFE, generalizable across any measurement $\boldsymbol{y}$.

## 2 PRELIMINARIES

### 2.1 SCORE-BASED DIFFUSION MODELS

We adopt the standard framework for constructing a continuous diffusion process $\boldsymbol{x}(t)$, where $t \in [0, T]$ and $\boldsymbol{x}(t) \in \mathbb{R}^d$, as outlined by Song et al. (2021c). Specifically, our goal is to initialize $\boldsymbol{x}(0)$ from a distribution $p_0(\boldsymbol{x}) = p_{\text{data}}$, and evolve $\boldsymbol{x}(t)$ towards a reference distribution $p_T$ at time $T$, which is easy to sample from.

The evolution of $\boldsymbol{x}(t)$ is governed by the Itô stochastic differential equation (SDE):

$$d\boldsymbol{x} = \boldsymbol{f}(\boldsymbol{x}, t)dt + g(t)d\boldsymbol{w}, \tag{3}$$

where $\boldsymbol{f} : \mathbb{R}^d \times \mathbb{R} \to \mathbb{R}^d$ represents the drift function, and $g : \mathbb{R} \to \mathbb{R}^d$ denotes the diffusion coefficient. These coefficients are designed to drive $\boldsymbol{x}(t)$ towards a spherical Gaussian distribution as $t$ approaches $T$. When the drift function $\boldsymbol{f}(\boldsymbol{x}, t)$ is affine, the perturbation kernel $p_{0t}(\boldsymbol{x}(t)|\boldsymbol{x}(0))$

is Gaussian, allowing for the parameters to be determined analytically. This facilitates data perturbation via $p_{0t}(\boldsymbol{x}(t)|\boldsymbol{x}(0))$ efficiently, without necessitating computations through a neural network.

Furthermore, corresponding to the forward SDE, there exists a reverse-time SDE:

$$d\boldsymbol{x} = [\boldsymbol{f}(\boldsymbol{x},t) - g(t)^2\nabla_{\boldsymbol{x}}\log p_t(\boldsymbol{x})]dt + g(t)d\bar{\boldsymbol{w}}, \qquad (4)$$

where $dt$ represents an infinitesimal negative time step, and $\bar{\boldsymbol{w}}$ denotes the backward standard Brownian motion. While the trajectory of Eq. (4) is stochastic, there also exists a corresponding probability-flow ODE (PF-ODE) that recovers the same law $p_t(\boldsymbol{x})$ as the time progresses (Song et al., 2021c;a)

$$d\boldsymbol{x} = [\boldsymbol{f}(\boldsymbol{x},t) - \frac{g(t)^2}{2}\nabla_{\boldsymbol{x}}\log p_t(\boldsymbol{x})]dt. \qquad (5)$$

This allows a deterministic mapping between the reference and the target distribution, and hence diffusion models can also be seen as a neural ODE (Chen et al., 2018).

A neural network can be trained to approximate the true score function $\nabla_{\boldsymbol{x}}\log p_t(\boldsymbol{x})$ through score matching techniques, as demonstrated in previous works (Song & Ermon, 2019; Song et al., 2021c). This approximation, denoted $\boldsymbol{s}_\theta(\boldsymbol{x},t) \approx \nabla_{\boldsymbol{x}}\log p_t(\boldsymbol{x})$, is then utilized to numerically integrate the reverse-time SDE. To effectively train the score function, denoising score matching (DSM) is often employed (Hyvärinen & Dayan, 2005)

$$\theta^* = \arg\min_\theta \mathbb{E}_{t\sim U(\varepsilon,1),\boldsymbol{x}(t),\boldsymbol{x}(0)}\left[\|\boldsymbol{s}_\theta(\boldsymbol{x}(t),t) - \nabla_{\boldsymbol{x}_t}\log p_{0t}(\boldsymbol{x}(t)|\boldsymbol{x}(0))\|_2^2\right], \qquad (6)$$

Interestingly, the posterior mean, or the so-called denoised estimate can be computed via Tweedie's formula (Efron, 2011). Specifically, for $p(\boldsymbol{x}_t|\boldsymbol{x}_0) = \mathcal{N}(\boldsymbol{x}_t; \alpha_t\boldsymbol{x}_0, \beta_t^2\boldsymbol{I})$,

$$\hat{\boldsymbol{x}}_{0|t}^\theta := \mathbb{E}_{p(\boldsymbol{x}_0|\boldsymbol{x}_t)}[\boldsymbol{x}_0|\boldsymbol{x}_t] = \frac{1}{\alpha_t}(\boldsymbol{x}_t + \beta_t^2\nabla_{\boldsymbol{x}_t}\log p(\boldsymbol{x}_t)). \qquad (7)$$

## 2.2 DIFFUSION MODELS FOR INVERSE PROBLEMS (DIS)

Solving the reverse SDE in Eq. (4) or the PF-ODE in Eq. (5) results in sampling from the prior distribution $p_\theta(\boldsymbol{x}_0)$, with the subscript emphasizing the time variable in the diffusion model context $\boldsymbol{x}_0 \equiv \boldsymbol{x}$. When solving an inverse problem as posed in Eq. (1), our goal is to sample from the posterior $p_\theta(\boldsymbol{x}_0|\boldsymbol{y}) \propto p_\theta(\boldsymbol{x})p(\boldsymbol{y}|\boldsymbol{x}_0)$. Using Bayes rule for a general timestep $t$ yields

$$\nabla_{\boldsymbol{x}_t}\log p_\theta(\boldsymbol{x}_t|\boldsymbol{y}) = \nabla_{\boldsymbol{x}_t}\log p_\theta(\boldsymbol{x}_t) + \nabla_{\boldsymbol{x}_t}\log p(\boldsymbol{y}|\boldsymbol{x}_t). \qquad (8)$$

While the former term can be replaced with a pre-trained diffusion model, the latter term is intractable and needs some form of approximation. Existing DIS (Kawar et al., 2022; Chung et al., 2023a; Wang et al., 2023) propose different approximations for $\nabla_{\boldsymbol{x}_t}\log p(\boldsymbol{y}|\boldsymbol{x}_t)$, which yields sampling from slightly different posteriors $\nabla_{\boldsymbol{x}_t}\log p_\theta(\boldsymbol{x}_t|\boldsymbol{y})$.

Algorithmically, the posterior samplers are often implemented so that the original numerical solver for sampling from the prior distribution remains intact, while modifying the Tweedie denoised estimate at each time $\hat{\boldsymbol{x}}_{0|t}^\theta$ to satisfy the measurement condition given as Eq. (1). From Tweedie's formula, we can see that this corresponds to approximating the conditional posterior mean $\mathbb{E}[\boldsymbol{x}_0|\boldsymbol{x}_t,\boldsymbol{y}]$ in the place of the unconditional counterpart $\mathbb{E}[\boldsymbol{x}_0|\boldsymbol{x}_t]$. The algorithms are inherently iterative, and the modern solvers (Chung et al., 2023a; Wang et al., 2023; Zhu et al., 2023) require at least 50 NFE to yield a high-quality sample. Moreover, as existing methods can be interpreted as approximating the reverse distribution $p(\boldsymbol{x}_0|\boldsymbol{x}_t)$ with a simplistic Gaussian distribution $q(\boldsymbol{x}_0|\boldsymbol{x}_t) = \mathcal{N}(\boldsymbol{x}_0; \hat{\boldsymbol{x}}_{0|t}^\theta, s_t^2\boldsymbol{I})$ (Peng et al., 2024), it typically yields a large approximation error, especially in the earlier steps of the reverse diffusion.

## 3 RELATED WORKS

### 3.1 VARIATIONAL INFERENCE IN DIS

Standard DIS discussed in Sec. 2.2 sample from the posterior distribution by following the reverse diffusion trajectory. Another less studied approach uses VI to use a new proposal distribution, where

| Class | Score-based | | Variational Inference | | |
|---|---|---|---|---|---|
| Methods | DIS | Noise2Score | RED-Diff | Score prior | **APS (ours)** |
| One-step inference | ✗ | ✓ | ✓ | ✓ | ✓ |
| Tackles general inverse problems | ✓ | ✗ | ✗ | ✓ | ✓ |
| Exact likelihood computation | ✗ | ✗ | ✗ | ✓ | ✓ |
| Amortized across $\boldsymbol{y}$ | ✗ | ✗ | ✗ | ✗ | ✓ |
| Generalizable across dataset | ✗ | ✗ | ✗ | ✗ | ✓ |
| Blind sampling | ✗ | ✗ | ✗ | ✗ | ✓ |

Table 1: Methods that leverage diffusion priors for solving inverse problems according to their class, and their characteristics.

the problem is cast as an optimization problem in Eq. (2). RED-diff (Mardani et al., 2023) places a unimodal Gaussian distribution as the proposal distribution $q_\phi^{\boldsymbol{y}}(\boldsymbol{x})$, and the KL minimization is done in a coarse-to-fine manner, similar to standard DIS, starting from high noise level to low noise level. While motivated differently, RED-diff and standard DIS have similar downsides of requiring at least a few tens of NFEs, as well as placing a simplistic proposal distribution. Furthermore, one can achieve only a single sample per optimization.

Recently, Feng *et al.* (Feng et al., 2023; Feng & Bouman, 2023) uses an NF model for the proposal distribution while solving the same VI problem. The optimization problem involves computing the diffusion prior log likelihood $\log p_\theta(\boldsymbol{x})$. It was shown that it can be exactly computed by solving the PF-ODE (Feng et al., 2023; Song et al., 2021c), but numerically solving the PF-ODE per every optimization step is extremely computationally heavy, and hence does not scale well. To circumvent this issue, it was proposed to use a lower bound (Feng & Bouman, 2023; Song et al., 2021b). Once trained, the NF model can be given different noise inputs $\boldsymbol{z} \sim \mathcal{N}(0, \boldsymbol{I})$ to generate diverse posterior samples with a single forward pass through the network. However, the training should be performed with respect to all the different measurements, not being able to generalize across the dataset. Our work follows along this path to overcome the current drawback and optimize a single model for entire measurement space with a similar cost as shown in Tab. 1.

## 3.2 DISTILLATION OF THE DIFFUSION PRIOR

Our method involves distillation of the diffusion prior into a student deep neural network, in our case an NF model. Particularly, it involves evaluating the output of the model by checking the denoising loss gradients from the pre-trained diffusion model. This idea is closely related to variants of score distillation sampling (SDS) (Poole et al., 2023; Wang et al., 2024), where the gradient from the denoising loss is used to distill the diffusion prior by discarding the score Jacobian. Possibly a closely related work is Diff-instruct (Luo et al., 2024), where the authors propose to train a one-step generative model similar to GANs (Goodfellow et al., 2014) by distillation of the diffusion prior with VI. By proposing an integral KL divergence (IKL) by considering KL minimization across multiple noise levels across the diffusion, it was shown that SDS-like gradients can be used to effectively train a new generative model. While having similarities, our method directly minimizes the KL divergence and does not require dropping the score Jacobian.

Orthogonal to the score distillation approaches, there have been recent efforts to train a student network to emulate the PF-ODE trajectory itself (Song et al., 2023b; Gu et al., 2023) with a single NFE, one of the most prominent directions being consistency distillation (CD) (Song et al., 2023b). While promising, the performance of CM is upper-bounded by the *teacher* PF-ODE. Thus, in order to leverage CD-type approaches for diffusion posterior sampling, one has to choose one of the approximations of DIS as its teacher model. In this regard, applying CD for diffusion inverse problem solving is inherently limited.

## 4  AMORTIZED POSTERIOR SAMPLING (APS)

### 4.1  CONDITIONAL NF FOR AMORTIZED SCORE PRIOR

The goal is to use a variational distribution that is conditioned on $\boldsymbol{y}$, such that the resulting distilled conditional NF model $G_\phi$ generalizes to any condition $\boldsymbol{y}$. To this end, inspired from the choices of (Sun & Bouman, 2021; Feng et al., 2023) we modify the objective in Eq. (2) to

$$\min_\phi D_{KL}(q_\phi(\boldsymbol{x}_0|\boldsymbol{y})||p_\theta(\boldsymbol{x}_0|\boldsymbol{y})) \tag{9}$$

$$= \min_\phi \int q_\phi(\boldsymbol{x}_0|\boldsymbol{y})[-\log p(\boldsymbol{y}|\boldsymbol{x}_0) - \log p_\theta(\boldsymbol{x}_0) + \log q_\phi(\boldsymbol{x}_0|\boldsymbol{y})] \tag{10}$$

$$= \min_\phi \mathbb{E}_{\boldsymbol{z}}\left[\underbrace{-\log p(\boldsymbol{y}|G_\phi(\boldsymbol{z},\boldsymbol{y}))}_{\text{fidelity}} - \underbrace{\log p_\theta(G_\phi(\boldsymbol{z},\boldsymbol{y}))}_{\text{prior}} + \underbrace{\log \pi(\boldsymbol{z}) - \log\left|\det\frac{dG_\phi(\boldsymbol{z},\boldsymbol{y})}{d\boldsymbol{z}}\right|}_{\text{induced entropy}}\right], \tag{11}$$

where the second equality is the result of choosing a conditional NF as our proposal distribution, and now the expectation is over random noise $\boldsymbol{z} \sim \mathcal{N}(0, \boldsymbol{I})$. Notice that our network takes in both a random noise $\boldsymbol{z}$ and the condition $\boldsymbol{y}$ as an input to the network.

Under the Gaussian measurement model in Eq. (1), the fidelity loss reads

$$-\mathbb{E}_{\boldsymbol{z}}[\log p(\boldsymbol{y}|G_\phi(\boldsymbol{z},\boldsymbol{y}))] = -\mathbb{E}_{\boldsymbol{z}}\left[\frac{\|\boldsymbol{y} - \mathcal{A}(G_\phi(\boldsymbol{z},\boldsymbol{y}))\|_2^2}{2\sigma_y^2}\right]. \tag{12}$$

Moreover, the induced entropy can be easily computed as it is an NF

$$\mathbb{E}_{q_\phi(\boldsymbol{x})}[\log q_\phi(\boldsymbol{x})] = \mathbb{E}_{\boldsymbol{z}}\left[\log \pi(\boldsymbol{z}) - \log\left|\det\frac{dG_\phi(\boldsymbol{z},\boldsymbol{y})}{d\boldsymbol{z}}\right|\right] \tag{13}$$

where $\pi(z)$, in our case, is the reference Gaussian distribution $\mathcal{N}$. For simplicity, let us denote $\hat{\boldsymbol{x}}_0 := G_\phi(\boldsymbol{z}, \boldsymbol{y})$.

Computation of $\log p_\theta(\hat{\boldsymbol{x}}_0)$ is more involved: to exactly compute the value, we would have to solve the PF-ODE, which is compute-heavy (Song et al., 2021c; Feng et al., 2023). To circumvent this burden, we leverage the evidence lower bound (ELBO) (Song et al., 2021b; Feng & Bouman, 2023) $b_\theta(\hat{\boldsymbol{x}}_0) \leq \log p_\theta(\hat{\boldsymbol{x}}_0)$:

$$b_\theta(\hat{\boldsymbol{x}}_0) = \mathbb{E}_{p(\hat{\boldsymbol{x}}_T|\hat{\boldsymbol{x}}_0)}[\log \pi(\hat{\boldsymbol{x}}_T)] - \frac{1}{2}\int_0^T g(t)^2 h(t)\, dt \tag{14}$$

where

$$h(t) := \mathbb{E}_{p(\hat{\boldsymbol{x}}_t|\hat{\boldsymbol{x}}_0)}\left[\underbrace{\|\boldsymbol{s}_\theta(\hat{\boldsymbol{x}}_t) - \nabla_{\hat{\boldsymbol{x}}_t}\log p(\hat{\boldsymbol{x}}_t|\hat{\boldsymbol{x}}_0)\|_2^2}_{\text{DSM(Eq. (6))}}\right.$$

$$\left. - \|\nabla_{\hat{\boldsymbol{x}}_t}\log p(\hat{\boldsymbol{x}}_t|\hat{\boldsymbol{x}}_0)\|_2^2 - \frac{2}{g(t)^2}\nabla_{\hat{\boldsymbol{x}}_t} \cdot \boldsymbol{f}(\hat{\boldsymbol{x}}_t, t).\right] \tag{15}$$

When we have $p(\boldsymbol{x}_t|\boldsymbol{x}_0) = \mathcal{N}(\boldsymbol{x}_t; \alpha_t\boldsymbol{x}_0, \beta_t^2\boldsymbol{I})$ and a standard diffusion model with a linear SDE $\boldsymbol{f}(\boldsymbol{x}_t, t) = f(t)\boldsymbol{x}_t$,

$$\|\nabla_{\boldsymbol{x}_t}\log p(\boldsymbol{x}_t|\boldsymbol{x}_0)\|_2^2 = \frac{1}{\beta(t)^2}\|\boldsymbol{\epsilon}\|_2^2, \quad \frac{2}{g(t)^2}\nabla_{\boldsymbol{x}_t} \cdot \boldsymbol{f}(\boldsymbol{x}_t, t) = \frac{2d\beta(t)}{g(t)^2}, \tag{16}$$

where $d$ is the dimensionality of $\boldsymbol{x}_t$, and both terms are independent of $\phi$ and $\theta$. Intuitively, the DSM term evaluates the probability of $\boldsymbol{x}_0$ by measuring how easy it is to denoise the given $\boldsymbol{x}_0$. When the network easily denoises the given image, then it will assign a high probability. When not, a low probability is assigned. We can now define an equivalent ELBO $b'_\theta(\boldsymbol{x}_0)$ in terms of optimization, which reads

$$b'_\theta(\boldsymbol{x}_0) = \mathbb{E}_{p_{0T}}[\log \pi(\boldsymbol{x}_T)] - \frac{1}{2}\int_0^T g(t)^2\|\boldsymbol{s}_\theta(\boldsymbol{x}_t) - \nabla_{\boldsymbol{x}_t}\log p(\boldsymbol{x}_t|\boldsymbol{x}_0)\|_2^2\, dt \tag{17}$$

Plugging $b'_\theta(G_\phi(\boldsymbol{z}, \boldsymbol{y}))$ of Eq. (17) in the place of $\log p_\theta(G_\phi(\boldsymbol{z}, \boldsymbol{y}))$ in Eq. (11), we can efficiently update $\phi$ by distilling the prior information contained in the diffusion model.

## 4.2 ARCHITECTURE

It has been demonstrated in (Lugmayr et al., 2020) that Conditional NFs are capable of learning distributions on the ambient space that are constrained on measurement. To achieve an architecture with invertible transformations, we extend RealNVP (Dinh et al., 2016) architecture to the conditional settings by borrowing insight from (Sun & Bouman, 2021). In its plain form, RealNVP architecture mainly consists of Flow steps, each containing two Affine Coupling layers. In each affine coupling layer, input signal $x$ is split into two parts: $x_a$ which stays unchanged, and $x_b$ which is fed into the neural network. In order to invoke the condition, we simply concatenate the conditioning input $y$ to the $x_b$ as these layers serve as the main and basic building blocks of entire invertible architecture. This seemingly simple integration led to very promising results in both Euclidean and non-Euclidean geometries as will be depicted in Section 5.

## 4.3 MANIFOLD

Many real-world datasets, particularly in environmental science, naturally reside on non-Euclidean geometries, making inverse problems challenging. Our work extends conditional normalizing flows (CNFs) to distributions on non-Euclidean manifolds, enabling direct solving of inverse problems on these surfaces without additional rendering steps. We represent manifold data as point clouds of size $V \times C$, where $V$ is the number of vertices in the mesh discretization and $C$ is the dimension of signal features. By leveraging the expressive power of CNFs, our approach captures the intrinsic geometry and structure of manifold data while enabling efficient inference and sampling. Our framework can handle complex geometries and severe masking levels across different manifolds, as demonstrated in our experiments with noisy inpainting and imputation tasks (see Section 5).

## 5 EXPERIMENTS

We validate our approach through various experiments, including (i) Denoising, Super Resolution (SR), and Deblurring with CelebA face image data (Liu et al., 2015); (ii) Inpainting on Stanford Bunny Manifold with MNIST data; and (iii) Imputation on Sphere with ERA5 (Hersbach et al., 2020) temperature data. (i) Denoising, SR, and Deblurring are performed on the Euclidean in the image domain. In contrast, noisy (ii) inpainting and (iii) imputation are solved directly on the bunny and sphere manifolds. Throughout all the experiments, we use 24 flow steps and we set the batch size to the 64. We conduct all the training and optimization experiments on a single RTX3090 GPU instance. Our code is implemented in the JAX framework (Bradbury et al., 2018).

### 5.1 EXPERIMENTAL SETTINGS

**Inverse Problems on CelebA.** We follow the usual formulation and adapt $32 \times 32$ resolution of facial images. Data is normalized into [0, 1] range and measurement is acquired by the appropriate choices of forward operator depending on the task (See Appendix A for details). $G_\phi$ is optimized over the $19,962$ **test** images by using the forward operator and prior from diffusion models. We optimize APS for 1M iterations for all different tasks (convergency was observed earlier but continued for potential refinement).

**Inpainting on Bunny MNIST.** In order to demonstrate the geometric awareness of our model, we conduct experiment on Stanford Bunny Manifold. We choose the mesh resolution of 1889 vertices and then project the [0, 1] normalized MNIST digits onto the bunny manifold (Turk & Levoy, 1994). In order to ensure the dimensionality compatibility for the models, we use 1888 vertices and zero mask the last vertex throughout the experiments. We obtain the measurement by occluding $30\%$ of vertices randomly and adding some Gaussian noise, i.e. $\mathcal{A}$ is the random masking operator and $\sigma_y = 0.1$ in (1). APS is optimized on the test chunk of $10,000$ digit examples for 1.5M iterations.

**Imputation on ERA5.** To show the essence and practical importance of our pipeline, we further conduct experiments on ERA5 temperature dataset. Even though data is available in a rectangular format, due to the spherical shape of Earth, it inherits some geometric information. We use $4°$ resolution dataset with 4140 vertices borrowed from (Dupont et al., 2022b) with only temperature channel as it is quite popular to analyze in the domain of generative AI (Dupont et al., 2021; 2022a). Again, due to the dimensionality, we add 20 more vertices with a signal value of zero, and the data

| Geometry | Euclidean | | | Riemannian | | | | | |
|---|---|---|---|---|---|---|---|---|---|
| Dataset & Task | celebA (denoising) | | | Bunny-MNIST (inpainting) | | | ERA5 (imputation) | | |
| Metric | Time[s]↓ | PSNR↑ | SSIM↑ | Time[s]↓ | PSNR↑ | MSE↓ | Time[s]↓ | PSNR↑ | SSIM↑ |
| MCG (Chung et al., 2022a) | - | - | - | 19.85 | 26.69 | 0.0024 | 16.16 | 27.52 | 0.871 |
| Noise2Score (Kim & Ye, 2021) | 0.0172 | 24.36 | 0.871 | - | - | - | - | - | - |
| DPS (Chung et al., 2023a) | 16.95 | **27.93** | **0.932** | 19.39 | **28.03** | **0.0017** | 15.36 | 28.95 | 0.953 |
| APS (ours) (N = 1) | **0.0021** | 23.37 | 0.836 | **0.0021** | 25.97 | 0.0032 | **0.0012** | 33.17 | 0.883 |
| APS (ours) (N = 128) | 0.0035 | 25.82 | 0.901 | 0.0035 | 26.72 | 0.0022 | 0.0018 | **34.61** | **0.959** |

Table 2: Quantitative results on our 3 main experiments. **Best**, second best

is [0, 1] normalized. In contrast to Bunny MNIST, we use more severe occlusion of $60\%$ random masking with additional Gaussian noise of $\sigma_y = 0.05$. We perform the optimization of APS on the test part of the dataset with 2420 examples for 315k iterations.

**Score Networks.** For all the diffusion priors, VPSDE formulation has been adapted. In the case of image domain CelebA, we borrow the same score checkpoint used in the (Feng et al., 2023; Feng & Bouman, 2023), which uses NCSN++ (Song et al., 2021b) architecture and has been trained for 1M iterations. For Bunny MNIST, we adapt the 1D formulation of DDPM (Ho et al., 2020) and train the score network for 500k, at which the convergence was clearly observed through the generated samples. In the case of spherical weather data, we followed the same strategy as Bunny MNIST but achieved convergence of score network earlier at 360k iterations.

## 5.2 RESULTS

In this section, we provide the general results of each different task described above. We compare APS with the various baselines including, DPS (Chung et al., 2023a), MCG (Chung et al., 2022a), and Noise2Score (Kim & Ye, 2021). It should be noted that MCG and DPS are identical for denoising, and Noise2Score is only applicable to denoising. In such cases, we do not report the metrics. We also demonstrate the comparisons and results against Feng *et al.* (Feng & Bouman, 2023). Finally, we experimentally confirm the robustness of APS across different unseen data or datasets. For evaluation purposes, we use peak signal-to-noise ratio (PSNR) and structural-similarity-index-measure (SSIM) which are widely used to assess the performance of inverse solvers with the ground truth and reconstructed signals. We further evaluate Fréchet Inception Distance (FID) to showcase the perceptual quality of generated samples. As the proposed method, APS, can sample multiple different posterior samples with a single forward pass, and this process is easily parallelizable, we report two different types for the proposed method. One by taking a single posterior sample ($N = 1$), and another by taking 128 posterior samples and taking the mean ($N = 128$).

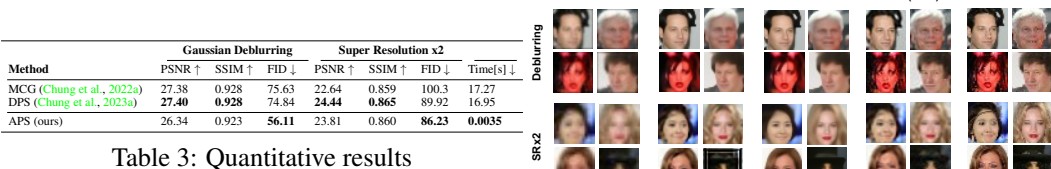

| Method | Gaussian Deblurring | | | Super Resolution x2 | | | |
|---|---|---|---|---|---|---|---|
| | PSNR↑ | SSIM↑ | FID↓ | PSNR↑ | SSIM↑ | FID↓ | Time[s]↓ |
| MCG (Chung et al., 2022a) | 27.38 | 0.928 | 75.63 | 22.64 | 0.859 | 100.3 | 17.27 |
| DPS (Chung et al., 2023a) | 27.40 | 0.928 | 74.84 | 24.44 | 0.865 | 89.92 | 16.95 |
| APS (ours) | 26.34 | 0.923 | **56.11** | 23.81 | 0.860 | **86.23** | 0.0035 |

Table 3: Quantitative results

Figure 3: Qualitative results

### 5.2.1 GENERAL RESULTS.

In general, our approach achieves competitive quantitative and qualitative results across different datasets on Euclidean and non-Euclidean geometries. We observe significant time improvements due to the single-step generation ability of our framework. Tab. 2 and Fig. 1,6 depict competitive quantitative and qualitative results confirming discussions along with instant time generations. It should be noted that the boosted version ($N = 128$) of the proposed method only marginally increases the compute time, as we can sample multiple reconstructions in parallel. To demonstrate that our method can be applied to more general inverse problems, similar to (Chung et al., 2023a), we

conducted 2× Super Resolution and Gaussian Deblurring experiments on the celebA data as shown in Tab. 3 & Fig. 3, where we see that the perceptual quality of the proposed method is *better* while being ∼ ×1000 faster, and the difference in the distortion metrics are small. Interestingly, while Noise2Score approximates the posterior mean, and the boosted version of the proposed method also approximates the posterior mean by taking the average of the posterior samples, our method outperforms Noise2Score by more than 1 db, showcasing the superiority of the proposed method.

### 5.2.2 COMPARING WITH DIS AND NOISE2SCORE.

Both DPS (Chung et al., 2023a) and MCG (Chung et al., 2022a) leverage the pre-trained diffusion model to sample from the posterior distribution. However, these methods require thousands of NFEs to achieve stable performance. The required time for DPS and MCG is reported in Tab. 2, 3. When decreasing the NFE as shown in Fig. 4 (a), PSNR heavily degrades and eventually diverges when we take an NFE value of less than 30. APS achieves competitive performance even with a single NFE. Moreover, it is shown in Fig. 4 (b) that even slightly incorrectly choosing the step size parameter leads to a large degradation in performance, whereas our method is free from such cumbersome hyperparameter tuning. Finally, it is shown in Fig. 4 (c) that DPS collapses to the mean of the prior distribution, altering the content of the measurement heavily when we take a smaller number of NFEs.

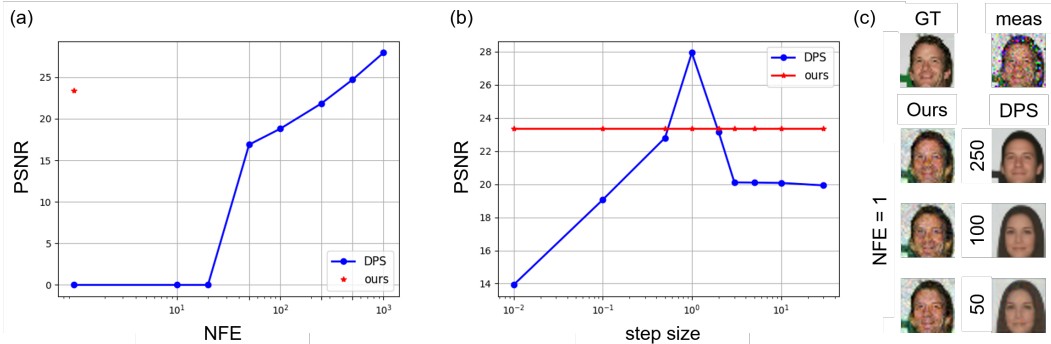

Figure 4: Comparison of our method against DPS (Chung et al., 2023a) on celebA denoising. (a) NFE vs. PSNR plot, (b) step size (used in DPS only) vs. PSNR plot, (c) representative results by varying the NFE.

It is worth mentioning that Noise2Score (Kim & Ye, 2021) is applicable for one-step denoising of the measurements by leveraging the Tweedie's formula. However, as discussed in Sec. 6, APS is generally applicable to a wide class of inverse problems, whereas the applicability of Noise2Score is limited.

### 5.2.3 COMPARING WITH SCORE PRIOR METHOD.

Compared to the exact score prior (Feng et al., 2023), surrogate counterpart (Feng & Bouman, 2023) presents 100 times faster approach along with competitive or slightly better results in terms of quality. Despite being fast in terms of optimization of NF, Feng & Bouman (2023) still requires training the network for a considerable amount of time for every single measurement. We observed that under same conditions, conditional NF does not increase the complexity and training stage takes 0.15 seconds which is 0.14 seconds in case of unconditional version. We further sample a random point from test data of celebA and optimize unconditional NF with the same configurations as ours on this single measurement. NF trained solely on this data reaches 23.75dB in PSNR score, which is almost same as our result of 23.43dB on this measurement. All these confirms that under same conditions, APS can simply achieve best results being also amortized for plenty of measurements.

### 5.2.4 GENERALIZABILITY ACROSS DATASETS AND BLIND INVERSE PROBLEMS.

We further observe that our optimized framework can be used on unseen data as well. Tab. 5 and Fig. 7 depicts that we achieve similar quantitative and qualitative reconstruction results when we sample from unseen celebA or ERA5 validation datasets. Note that score network is trained on train

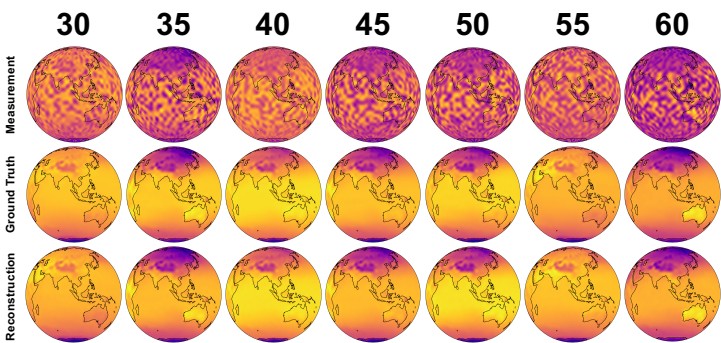

Figure 5: Blind inverse problem solving with varying imputation levels.

signals and CNF optimization has been conducted on test signals, i.e. validation is totally hidden to both teacher and student models. More strongly, our approach can also be leveraged on various datasets as pre-optimized inverse solver. To this end, we use our celebA optimized CNF model to perform Denoising task on the FFHQ (Karras et al., 2019) dataset. Same Table and Figure show that our model can remove the noise artifacts with a similar performance as it does on original data, confirming the generalizibility feature.

We further observed that APS can work in the absence of forward operator. In other words, we can perform blind inverse problems through our amortized posterior sampling. We used various imputation levels between 30% to 60% for ERA5 dataset, and conducted experiments with random choice of imputation in a blind manner. As a result, Fig. 5 shows that results as good as the original inverse solver with the known forward operation (at least 33 PSNR across all different blind imputation levels).

## 6 DISCUSSION

We show a first proof of concept that we can construct a one-step posterior sampler that generalizes across any measurements in an unsupervised fashion (only having access to the measurements $y$). Notably, APS extends to wide use cases with minimal constraints: 1) the operator $\mathcal{A}$ can be arbitrarily complex and non-linear, as in DPS (Chung et al., 2023a), unlike many recent DIS that requires linearity of the operator (Wang et al., 2023; Chung et al., 2024; Zhu et al., 2023); 2) training of the sampler can be done without any strict conditions on the measurement, unlike recent unsupervised score training methods that require i.i.d. measurement conditions with the same randomized forward operator (Daras et al., 2023; Kawar et al., 2023); 3) method can be generalized into different geometries and datasets in a blind manner, unlike recent DIS methods require to know forward operator during sampling (Chung et al., 2023b; Mardani et al., 2023). We opted for simplicity in the architecture design of $G_\phi$, and avoided introducing inductive bias of spatial information by taking a vectorized input, potentially explaining the slight background noise in the reconstructions. Further optimization in the choice of network architecture is left as a future direction of study.

## 7 CONCLUSION

In this work, we propose to use a conditional NF for a VI-based optimization strategy to train a one-step posterior sampler, which implicitly samples from the posterior distribution defined from the pre-trained diffusion prior. We show that APS is highly generalizable, being able to reconstruct samples that are not seen during training, applicable to diverse forward measurements, and types of data, encompassing standard Euclidean geometry as well as data on general Riemannian manifolds. We believe that our work can act as a cornerstone for developing a fast, practical posterior sampler that distills the diffusion prior.

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

# A APPENDIX

## A.1 REPRODUCIBILITY AND DETAILS OF PARAMETERS

We further provide all the necessary details to replicate the results with our proposed method. Tab. 4 demonstrates the details of tasks and datasets along with the parameter choices for both prior score network training and Conditional NF optimization. Note that, we have validated our approach through 3 different inverse problems on the image dataset (celebA), where noise level was set to 0.1 for denoising and 0.01 for Super-Resolution and Gaussian Deblurring.

| Parameter | CelebA | Bunny | Sphere |
|---|---|---|---|
| resolution (#vertices) | $32 \times 32$ | 1889 | 4140 |
| distribution on manifold | - | MNIST | ERA5 |
| task | varying | Inpainting | Imputation |
| mask level | - | 30% | 60% |
| noise level | varying | 0.1 | 0.05 |
| #channels | 3 | 1 | 1 |
| normalized range | [0, 1] | [0, 1] | [0, 1] |
| (S) #train data | 162,770 | 60,000 | 8,510 |
| (S) batch size | 128 | 64 | 64 |
| (S) learning rate | 2e-4 | 2e-4 | 2e-4 |
| (S) #training iters | 1M | 500k | 360k |
| (C) #test data | 19,962 | 10,000 | 2,420 |
| (C) batch size | 64 | 64 | 64 |
| (C) learning rate | 1e-5 | 1e-5 | 1e-5 |
| (C) #optimization iters | 1M | 1.5M | 315k |

Table 4: Different configurations of hyperparameter choices for varying datasets and manifolds learned by APS. (S) and (C) denotes the parameter choices for score network and CNF optimization, respectively.

## A.2 QUALITATIVE COMPARISONS WITH BASELINES

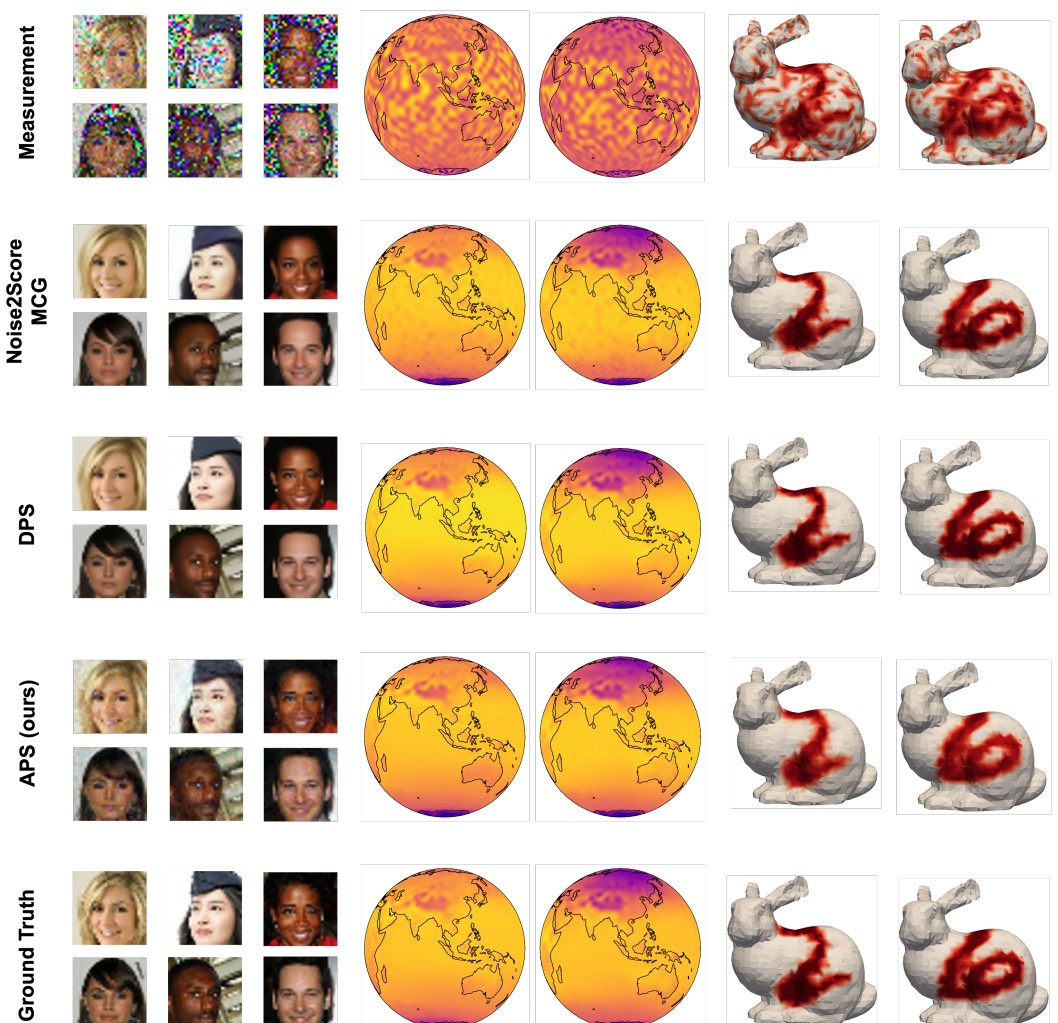

Figure 6: Comparisons with different baselines for CelebA denoising, ERA5 imputation, and Bunny MNIST inpainting tasks. Note that second row shows the results of Noise2Score for the CelebA denoising task and MCG for inpainting and imputation of manifold data.

## A.3 ROBUSTNESS RESULTS

| Dataset & Task | celebA val (denoising) | ERA5 val (imputation) | FFHQ (denoising) |
|---|---|---|---|
| PSNR↑ | 23.26 | 33.12 | 21.92 |
| SSIM↑ | 0.831 | 0.882 | 0.822 |

Table 5: APS is robust against unseen data samples and even generalizable accross different datasets once it is optimized. For the first 2 columns, we use validation part of datasets and feed-forward our CNF on this totally unseen data. For the third column, we even show that pre-optimized APS can be leveraged to restore back the noised data samples from across various datasets. All the quantitative results align with their counterparts in Tab. 2 that confirms robustness and generalization ability of our pipeline which was not possible before.

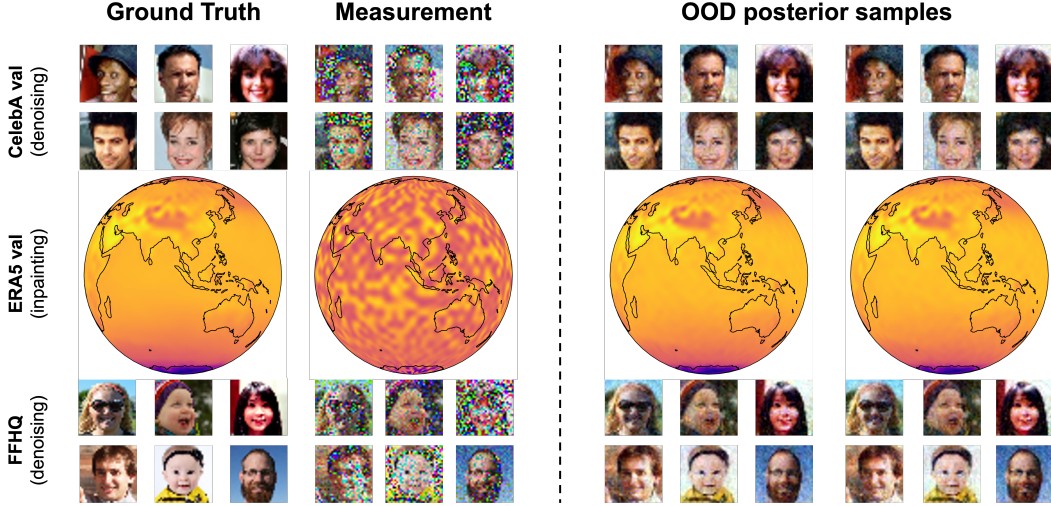

Figure 7: Figure demonstrating the visual results of robustness and generalization ability of APS. First and second rows show the results on unseen validation data, and third row depicts generalization to another dataset. Corresponding quantitative analysis can be found in Tab. 5.

