# OpenReview forum: "Amortized Posterior Sampling with Diffusion Prior Distillation"
_ICLR.cc/2025/Conference — ICLR 2025 Conference Withdrawn Submission_

### Official Review · Reviewer_oR42 · 2024-10-28

**Soundness:** 3
**Presentation:** 2
**Contribution:** 2
**Rating:** 3
**Confidence:** 4

**Summary:**

This paper presents APS, an efficient variational inference-based posterior sampling method that combines conditional normalizing flows with diffusion models to achieve one-step posterior samples. APS is unsupervised, does not require paired training data, and is capable of tackling inverse problems in both Euclidean and non-Euclidean spaces. Experimentally, APS is evaluated on image denoising, inpainting, and data imputation tasks, demonstrating speed improvements while maintaining competitive accuracy.

**Strengths:**

The method allows one-step inference for posterior sampling, offering substantial time savings over traditional DIS methods

APS is tested on both Euclidean and non-Euclidean data, showcasing its potential in various fields.

The integration of conditional NFs with diffusion priors, particularly in unsupervised settings, is promising.

**Weaknesses:**

Limited to specific models, such as MCG, DPS, and Noise2Score, but does not benchmark against other efficient variational methods recently introduced for inverse problems. Also benchmarking against similar methods would help demonstrate the behavior of this approach to others: DDNM, Pi-GDM, FPS, FPS-SMC, ect. For example, it’s unclear how the model can fail and what the failure cases may look like.

**Questions:**

Have the authors considered testing APS on broader inverse problem domains, beyond image and environmental data?

Could the authors clarify the performance of APS in terms of generalizability to entirely unseen problem classes?

Given APS’s unsupervised nature, how does it perform on ill-posed inverse problems where the prior distribution diverges significantly from the true posterior?

Is the observed background noise in reconstructions specific to the choice of vectorized input representation, or might this indicate a broader limitation of APS?

---

### Official Review · Reviewer_KEbV · 2024-10-31

**Soundness:** 3
**Presentation:** 3
**Contribution:** 2
**Rating:** 3
**Confidence:** 3

**Summary:**

This paper introduces Amortized Posterior Sampling (APS), a novel unsupervised variational inference approach for efficient posterior sampling in inverse problems using a conditional flow model. APS enables fast, diverse posterior sampling with a single neural evaluation and adapts to both Euclidean and non-Euclidean data. It achieves superior computational efficiency across tasks like image restoration and climate data imputation, providing real-time, high-quality solutions to various inverse problems.

**Strengths:**

The paper is well-written and easy to follow. It employs a conditional normalizing flow model to achieve one-step posterior inference. Experiments on both Euclidean and non-Euclidean datasets across various tasks—including inpainting, denoising, Gaussian deblurring, and super-resolution—demonstrate the proposed distillation method's fast speed and competitive performance.

**Weaknesses:**

1. Though the proposed method enables one-step inference, training such a model using equation (11) in the main text is time-consuming. Additionally, it requires training on test images, whereas previous methods like DPS perform zero-shot inference on test images and achieve strong performance. Given the strong performance in the field of diffusion-based inverse problems, I find APS's performance underwhelming. Furthermore, including more challenging tasks on higher-resolution images would be beneficial.

2. Although the paper highlights its novelty in variational inference, it lacks comparisons with relevant baselines. Methods like RED-DIFF and the score prior have not been included, likely because they are not distillation methods. More distillation-based approaches for posterior sampling should be considered.

3. The method appears to combine the reconstruction step and the normalizing-flow step in the paper, score prior,  into a single step, which limits its novelty.

**Questions:**

Please see the weakness for some of my concerns.

1. Can you clarify how many test images you use to obtain the metrics like PSNR and SSIM?

2. The use of varying metrics is a bit confusing; sometimes you report MSE, and other times FID or SSIM. A unified framework of metrics would be helpful.

---

### Official Review · Reviewer_KaqV · 2024-11-01

**Soundness:** 3
**Presentation:** 2
**Contribution:** 1
**Rating:** 3
**Confidence:** 5

**Summary:**

This paper introduces Amortized Posterior Sampling (APS), a variational inference method designed for efficient posterior sampling in inverse problems. APS trains a conditional flow model to align the variational distribution with the posterior distribution defined by a diffusion model, enabling diverse posterior samples with a single neural function evaluation. The authors demonstrate APS’s effectiveness across tasks like image restoration, manifold signal reconstruction, and climate data imputation, showing it significantly improves computational efficiency while maintaining competitive reconstruction quality for real-time applications across various fields.

**Strengths:**

- Clarity: The paper is clearly written and well-structured, making the methodology and key contributions easy to understand.

- Significance: Previous approaches to solving inverse problems often require multiple neural network evaluations, leading to high computational costs. This paper’s proposed method addresses this limitation by using an amortized variational inference framework, enabling efficient, single-step sampling. This improvement in computational efficiency represents a meaningful step forward for practical applications of inverse problem-solving.

**Weaknesses:**

- Limited novelty: The primary contribution of the paper is the use of amortization and diffusion model distillation to enable efficient, single-step sampling. However, this concept has already been explored in recent work [1] with a very similar approach, also leveraging **unsupervised diffusion prior distillation** with **amortized variational inference** for inverse problems. This challenges the paper's claim of novelty (i.e., "the first diffusion prior distillation" in line 101 or "the first proof of concept" in line 515). This highly relevant work is neither discussed in the paper nor compared experimentally. Thorough discussion and performance comparison with this prior work to highlight meaningful distinctions would strengthen the methodological contribution.

- Several existing frameworks, such as RED-Diff, also utilize variational inference objectives for inverse problem-solving. The primary distinction between RED-Diff and APS seems to be the replacement of the posterior distribution with a neural network conditioned on y; the diffusion prior loss remains similar. While the amortization of neural networks does improve efficiency, both methods are based on similar principles. Thus, an experimental comparison between RED-Diff and APS could highlight potential trade-offs in accuracy and generalization, providing a clearer picture of APS’s advantages.

- Limited performance and baselines: The paper lacks experimental comparisons with key baseline methods [1-5] that are also designed to solve inverse problems. For instance, while RED-Diff [2] is mentioned in Table 1 as a related framework, there is no direct comparison of its performance with APS. This gap in baseline evaluations weakens the empirical foundation of the paper, making it challenging to assess the benefits of APS. Additionally, the experimental results of APS do not seem strong in table 2 and qualitatively in Figure 6; artifacts and noise are still noticeable, even with limited baselines.

[1] Diffusion Prior-Based Amortized Variational Inference for Noisy Inverse Problems, ECCV24 \
[2] A variational perspective on solving inverse problems with diffusion models, ICLR23 \
[3] Pseudoinverse-guided diffusion models for inverse problems, ICLR23 \
[4] Zero-shot image restoration using denoising diffusion null-space model, ICLR23 \
[5] Denoising diffusion models for plug-and-play image restoration, CVPRW23

**Questions:**

- Please clarify the points raised in the weaknesses section.

- Is the neural network for the variational distribution optimized separately for each task, or can a single network generalize across multiple inverse problem tasks?

- The ability of APS to generalize to out-of-distribution datasets (e.g., applying a CelebA-trained model to the FFHQ dataset) is an appealing property. Can the authors compare these results to relevant baselines? Recent work, such as [1], has demonstrated similar generalization capabilities, and a direct comparison would provide useful insights.

---

### Official Review · Reviewer_wiGW · 2024-11-02

**Soundness:** 3
**Presentation:** 3
**Contribution:** 2
**Rating:** 5
**Confidence:** 4

**Summary:**

The paper proposes to accelerate inverse problem-solving with diffusion priors. This is achieved by distilling the unconditional diffusion prior for a given set of inverse problems into a conditional normalizing flow model $q_{\phi}(\boldsymbol{x}|\boldsymbol{y})$ that can (approximately) sample from the posterior distribution $p(\boldsymbol{x}|\boldsymbol{y})$ given any input $\boldsymbol{y}$ with a single NFE. Training is performed with a principled variational objective that is mathematically sound, and experiments are performed on multiple data domains (euclidean/non-euclidean) showing $\approx$comparable results with significantly faster inference time.

**Strengths:**

- The paper is very well written and mathematically sound.
- The results cover diverse settings and show clear favorable runtime for APS.
- Related work is addressed well and the paper’s contribution is put in proper context.

**Weaknesses:**

- The method requires training at test time (e.g. on the test set). This stands in contradiction to the speed claim of the method being a fast sampler. The question then becomes: How large should the test set be to compensate for this prolonged training?
- The method is claimed to produce high-quality and diverse posterior samples. Nonetheless, the resulting samples look rather noisy (e.g. CelebA denoising in Fig. 1, row 1), and they seem to have little to no diversity.
- The datasets used in the experiments are rather unorthodox and make it hard to properly appreciate the performance of the method. For example, 32x32 CelebA is tiny \- it is hard to notice any meaningful differences at this resolution in the resulting samples. Similarly, the non-euclidean datasets seem to be relatively simple and quite constrained.

The main reason for my negative score is the relatively limited experiments and my current lack of understanding of the test scenario where APS could be advantageous. Nonetheless, the paper is well structured and the overall idea is interesting. I'm willing to raise my score if I'm provided convincing answers to these two critical issues.

A few caught minor typos:
- L093 \- **one** the Euclidean manifold \-\> **on** the Euclidean manifold
- L356 \- we set the batch size to **the** 64 \-\> we set the batch size to 64
- L364 \- **convergency**  \-\> **convergence**
- L377 \- “due to dimensionality”?
- L467 \- leveraging **the** Tweedie’s formula \-\> leveraging Tweedie’s formula
- L523 \- unlike recent DIS methods require \-\> unlike recent DIS methods **which** require

**Questions:**

- The method seems to require a rather large test set to learn a proper approximation of the conditional normalizing flow. How does the size of the test set affect the performance of APS?
- Do you think it is mainly the architecture of $G\_{\\phi}(z,y)$ that resulted in the reduced sample quality or is there a more fundamental reason related perhaps to the ELBO on $\\log p\_{\\theta}(\\hat{x}\_0)$?
- Did you benchmark the diversity of the samples from APS? For example, against the baseline Score-Prior by Feng et al. (2023)? Or even against “standard” zero-shot diffusion posterior samplers such as DPS/DDNM?
- Does APS lose the great advantage of zero-shot DIS methods being adaptable to any operator at test time without additional training?
- Is there an inherent limitation with respect to the resolution of images that APS can handle? If so, it is recommended to mention this explicitly in your discussion. If not, then perhaps APS can be applied to larger images (e.g. 128x128) for better benchmarking?
- The distortion (e.g. PSNR/SSIM) of APS can be improved by sampling N=128 posterior samples and averaging them to approximate the posterior mean. However, shouldn’t a fair comparison do the same for the baselines?

---

### Note · Authors · 2024-11-15

**Comment:**

I have read and agree with the venue's withdrawal policy on behalf of myself and my co-authors.

**Withdrawal Confirmation:**

I have read and agree with the venue's withdrawal policy on behalf of myself and my co-authors.